# The Differences in the Whole-Brain Functional Network between Cantonese-Mandarin Bilinguals and Mandarin Monolinguals

**DOI:** 10.3390/brainsci11030310

**Published:** 2021-03-02

**Authors:** Xiaoxuan Fan, Yujia Wu, Lei Cai, Jingwen Ma, Ning Pan, Xiaoyu Xu, Tao Sun, Jin Jing, Xiuhong Li

**Affiliations:** Department of Maternal and Child Health, School of Public Health, Sun Yat-Sen University, Guangzhou 510080, Guangdong, China; fanxx5@mail2.sysu.edu.cn (X.F.); wuyj49@mail3.sysu.edu.cn (Y.W.); cailei3@mail2.sysu.edu.cn (L.C.); majw3@mail2.sysu.edu.cn (J.M.); pann3@mail2.sysu.edu.cn (N.P.); xuxy39@mail2.sysu.edu.cn (X.X.); sunt8@mail2.sysu.edu.cn (T.S.); jingjin@mail.sysu.edu.cn (J.J.)

**Keywords:** Cantonese-Mandarin bilinguals, Mandarin monolinguals, resting state fMRI, network-based statistics (NBS), graph theory, functional network connectivity

## Abstract

Cantonese-Mandarin bilinguals are logographic-logographic bilinguals that provide a unique population for bilingual studies. Whole brain functional connectivity analysis makes up for the deficiencies of previous bilingual studies on the seed-based approach and helps give a complete picture of the brain connectivity profiles of logographic-logographic bilinguals. The current study is to explore the effect of the long-term logographic-logographic bilingual experience on the functional connectivity of the whole-brain network. Thirty Cantonese-Mandarin bilingual and 30 Mandarin monolingual college students were recruited in the study. Resting state functional magnetic resonance imaging (rs-fMRI) was performed to investigate the whole-brain functional connectivity differences by network-based statistics (NBS), and the differences in network efficiency were investigated by graph theory between the two groups (false discovery rate corrected for multiple comparisons, *q* = 0.05). Compared with the Mandarin monolingual group, Cantonese-Mandarin bilinguals increased functional connectivity between the bilateral frontoparietal and temporal regions and decreased functional connectivity in the bilateral occipital cortex and between the right sensorimotor region and bilateral prefrontal cortex. No significant differences in network efficiency were found between the two groups. Compared with the Mandarin monolinguals, Cantonese-Mandarin bilinguals had no significant discrepancies in network efficiency. However, the Cantonese-Mandarin bilinguals developed a more strongly connected subnetwork related to language control, inhibition, phonological and semantic processing, and memory retrieval, whereas a weaker connected subnetwork related to visual and phonology processing, and speech production also developed.

## 1. Introduction

With the development of globalization, an increasing number of people have become long-term bilingual (BG) individuals. Researchers believe that long-term BG experience results in changes in the brain’s functional network, as the brain is a highly adaptive system [1,2,3,4]. It is well known that language processing requires cooperation among multiple brain regions, which depends on functional connectivity among different brain regions [5]. This synchronization between anatomically distinct brain regions might be equally important or even more important than the amplitude of activation in single regions for behavior and cognition [6]. Resting state functional magnetic resonance imaging (rs-fMRI) can detect the coherent activity of related regions that are task-independent and thus provides an effective way to explore the long-term influence of BG experiences on the brain’s functional networks [7,8].

To date, most rs-fMRI studies of BGs have used a seed-based approach to assess changes in functional connectivity patterns. Luk et al. [9] chose the bilateral inferior frontal gyrus (IFG) as yjr region of interest (ROI), which is important for BG language switching, and found that there was stronger long-range frontal-occipital and frontal-parietal functional connectivity in alphabetic BGs and stronger intrafrontal connectivity in monolinguals (MGs). By including the posterior cingulate cortex (PCC) as a seed for the default mode network (DMN) and the anterior insula–frontal operculum (aIFO) as a seed for cognitive control networks, Grady et al. [10] found stronger functional connectivity in the frontoparietal control network (FPN) and DMN in alphabetic BGs than in MGs. In another study, the dorsal anterior cingulate cortex (dACC) and the left caudate nucleus (LCN) were chosen as language control ROIs, and the bilateral superior temporal gyrus (STG) and bilateral rolandic operculum (RolOp) were chosen as the language-processing ROIs. The results showed that the functional connectivity between the dACC and the left STG and between the dACC and the RolOp decreased in bimodal BGs compared with MGs [11]. Generally, the current functional connectivity analysis focused only on seed regions, mainly limited to language switching and control, and thus failed to give a complete picture of the brain connectivity profile.

In fact, numerous studies have found that language processing involves regions beyond the classic reading network, such as low-level visual regions, high-level executive function regions, and limbic and subcortical regions [12,13,14]. Therefore, whole-brain analysis might be a more informative method for understanding the effect of long-term BG experience on brain functional networks. Graph theory, a method extensively used recently, makes it possible to understand functional connectivity in a large-scale network. In addition, network-based statistics (NBS) analysis can identify components formed by a series of connections of nodes that were significantly different between the two groups, which can be called subnetworks instead of merely pairs of regions, thus providing information about the whole-brain functional organization [15]. Garcia-Penton et al. [16] investigated the whole-brain structural network of BGs, and found that, compared with MGs, two structural subnetworks related to language processing and monitoring had stronger connectivity in BGs. In addition, compared with BGs, MGs had higher global efficiency of the whole-brain structural network [16]. Although brain structural connectivity has been shown to be a physical substrate of resting-state functional connectivity [17,18,19], studies have shown that structural and functional networks are not completely consistent [17,20]. Unfortunately, the manner in which long-term bilingual experience affects the whole-brain functional network remains unclear, and it needs to be further clarified.

Moreover, previous studies on BGs [9,10] mainly used alphabetic languages. Several differences in orthography, phonology, semantics, and syntax have been found between alphabetic and logographic languages [21]. Meta-analyses of fMRI studies have also showed differences in brain activation when processing alphabetic and logographic languages [22,23]. Chinese is a logographic language with complex visual-spatial structure, which requires more holistic visual analysis and map orthography to phonology at the whole character level than an alphabetic language [23]. In addition, some researchers believed that a Chinese character was a kind of whole-brain character, and the processing of a Chinese character requires the participation of a wide range of brain regions, such as phonological processing, visual processing, and the perception, motion, and executive control brain regions [23,24,25]. Unfortunately, up to now, there has been no whole-brain functional connectivity research on logographic BGs. Our previous study showed that the brain mechanisms of bilingual phonological processing in Cantonese-Mandarin bilinguals were different from that of alphabetic bilinguals [25]. Therefore, we speculate that logographic BGs would have different whole-brain functional networks compared with MGs.

Cantonese and Mandarin, as two major Chinese languages, are both logographic languages. The two languages share the same writing system, but they differ in lexicons, grammar, and especially phonology. The difference in pronunciation and rhyme between Cantonese and Mandarin is more than 76%; thus, a person with knowledge of the two languages is generally regarded as BG [26,27]. Therefore, Cantonese-Mandarin BGs provide a unique population to explore the effect of logographic BG experience on the whole-brain functional network. In mainland China, Cantonese mainly serves as a spoken language acquired and used by listening and speaking, but it is seldom used for reading and writing since Mandarin is the official language. In the current study, Cantonese-Mandarin BGs acquired the two languages before the age of 7 years old and were proficient in both languages, so we hypothesize that the Cantonese-Mandarin BGs might have a significant difference in the brain network in terms of phonological and semantic processing, visual processing, and perception, motion, and language control compared with MGs.

In order to test the above hypothesis, we used NBS analysis [15] to investigate the differences in whole-brain functional connectivity between Cantonese-Mandarin BGs and Mandarin MGs based on rs-fMRI. In addition, to assess the configuration properties of the whole-brain functional network, we employed graph theory analysis [28,29], which allowed us to explore differences in network efficiency between BGs and MGs in terms of quantitative parameters. Our results will provide new insight into how the brain’s functional network reorganizes to adapt to BG management.

## 2. Materials and Methods

### 2.1. Participants

Thirty-one Cantonese-Mandarin BGs and 31 Mandarin MGs were recruited in Guangzhou, Guangdong Province, China. All the participants in this study were undergraduates and postgraduates. After fMRI scanning, one MG participant was discarded because the scanned image was blurred, and one BG participant was discarded due to the maximum value of the head translation (rotation) of their functional brain data being over 3 mm (3°). Finally, 30 Mandarin MGs and 30 Cantonese-Mandarin BGs who learned Mandarin as their L2 between the ages of 3 and 7 were included in analysis. The Cantonese and Mandarin proficiency of all participants was evaluated according to the Language and Social Background Questionnaire (LBSQ). Developed by York University, the LBSQ has been proven to be valid and reliable for diverse language proficiency assessments by self-reporting and self-assessment [30]. To assess the proficiency of speaking, writing, understanding, and reading in Cantonese and Mandarin, a self-rating scale from 0 to 10, representing the lowest to highest proficiency, was applied. All BG participants were determined to be proficient in both languages (Table 1). In addition, since English education is universal in China, all participants in this study had English learning experience. To minimize the potential confounding effects, we matched the English proficiencies of the two groups by recruiting subjects who had passed the College English Test Band 4 (CET-4, a national English level test in China). We also measured the participants’ non-verbal intelligence quotient (IQ) with Raven’s Standard Progressive Matrices Test, given the link between non-verbal IQ and brain organization [31,32,33,34].

All subjects were right-handed according to the Edinburgh Handedness Inventory (EHI) [35], and none of them had experienced a head attack nor had any hearing impairment or neurological disorders. All participants signed the informed consent form before the study. The study was approved by the medical ethics committee of Sun Yat-sen University, and the ethic approval code is [L2016] No.036.

### 2.2. Data Acquisition

During rs-fMRI scanning, the subjects were asked to stay still, remain awake, and keep their eyes closed while not thinking about anything. All MRI datasets were acquired through a SIEMENS TRIO 3-T MRI scanner with the use of a 12 channel, phased-array, receiver-only head coil at the Center for the Study of Applied Psychology at the School of Psychology in South China Normal University. The resting-state data was collected in the same participants that were recruited for the project in our previous study [25]. In this project, resting-state fMRI data acquisition was collected before task-based fMRI during the same MRI session. For each participant, 240 functional images were obtained using weighted, single-shot echo planar imaging, and the parameters were set as follows: repetition time (TR) = 2000 ms, echo time (TE) = 30 ms, slice thickness = 3.5 mm, voxel size = 3.5 × 3.5 × 3.5 mm, flip angle = 90°, matrix size = 64 × 64, field of view (FOV) = 224 mm, number of slices = 32, and the slices were acquired interleaved. The 256 T1-weighted 3D images were acquired using magnetization-prepared rapid gradient-echo, and the parameters were set as TR = 1900 ms, TE = 2.52 ms, slice thickness = 1.0 mm, voxel size = 1.0 × 1.0 × 1.0 mm, flip angle = 9°, matrix size = 256 × 256, FOV = 256 mm, and number of slices = 176.

### 2.3. Data Preprocessing

Data preprocessing was performed using GRETNA (Version 2.0.0, http://www.nitrc.org/projects/gretna/ (accessed on 30 October 2020)) based on MATLAB R2016a. Before preprocessing, the first 10 volumes of each subject were discarded, considering adaptation to the environment. Then, we performed slice timing for the remaining 230 volumes to determine the differences among slices and realigned them to the first volume to correct the inevitable head movement during scanning. The subject was discarded if the maximum value of the head translation (rotation) of their functional brain data was over 3 mm (3°). Statistical analysis showed no significant differences between the BG and MG groups in head motion (*p* > 0.082 in any direction). Next, individual functional images were normalized to the Montreal Neurological Institute (MNI) space using the T1 structural data and resampled to 3 × 3 × 3 mm^3^ isotropic voxels. Finally, we performed linear detrending and bandpass filtering (0.01–0.08 Hz) and removed several nuisance signals from each voxel’s time series to reduce the effects of nonneuronal fluctuations, including head motion profiles from the Friston 24 parameters, the white matter signals, and the cerebrospinal fluid signal. Owing to the controversy of regressing out global signals in rs-fMRI [36,37], our study did not regress out the global signal.

### 2.4. Brain Network Construction

Based on graph theory, the human brain network was modeled as a graph, and the undirected weighted network of the whole brain was created for each participant. First, we constructed the brain functional network for each subject based on the Automated Anatomical Labeling (AAL) atlas template [38]. The template consists of 90 ROIs (45 for each hemisphere) (Appendix A) and is widely used in brain network studies [39,40,41], thus improving comparability across studies [42]. A node was used to represent a region, and using this configuration, we could then extract the time series of the 90 regions. Second, Pearson’s correlation coefficient *r* between each brain region was calculated as the strength of the functional connectivity between the pair of brain regions according to the extracted time series, so a 90 × 90 correlation coefficient matrix was obtained. To test whether functional connectivity was significantly different between the BG and MG groups, correlation coefficients were further converted into *z* values by using Fisher’s *r*-to-*z* transformation. This transformation generated values that were approximately normally distributed, and a *Z* statistic was then used to compare these transformed *z* values to determine the significance of the between-group differences in the correlations [43]. The functional connectivity between nodes i and j was considered as an edge, and the positive values were retained as weighted network connectivity.

### 2.5. Network-Based Statistic (NBS) Analysis

First, to reduce spurious correlations, a one-sample t-test was performed for every correlation in each group. We retained those correlations whose corresponding *p* values passed a statistical threshold of *p* < 0.05 after Bonferroni correction for each correlation matrix; otherwise, the correlations were set to zero. In addition, we obtained a union mask containing connectivity that was significant in either of the two groups. Then, we performed two-sample one-tailed t-tests (after controlling for the effects of age and sex) within this mask to determine group differences in functional connectivity, similar to previous studies [39,44,45]. A primary threshold (*p* < 1 × 10^−4^) was applied to identify suprathreshold links, among which the connected components and their size (the number of links included in the components) were detected. To estimate the significance of each component, a nonparametric permutation approach was used to derive the null distribution of the size of the connected component (5000 permutations). Briefly, all subjects were randomly reallocated into two groups in each permutation, and two-sample one-tailed t-tests were performed for the same set of links as mentioned above. The same primary threshold (*p* < 1 × 10^−4^) was subsequently used to produce suprathreshold links, within which the size of the maximal connected component was identified. Finally, for the connected component of size M identified in the real grouping of BGs and MGs, we calculated the proportion of the 5000 permutations, for which the maximal connected component was larger than M, to determine the corrected *p* value.

### 2.6. Graph Analysis

Graph theory was used to characterize the topological properties of the human brain functional networks [28,29]. As there would be different connection densities of graphs when applying different sparsity thresholds, we tested 36 sparsity thresholds ranging from 5% to 40% with an increase of 1%—as in a previous study [46]—to ensure the stability of network properties [47].

We computed five global network metrics, including the clustering coefficient (*Cp*), characteristic path length (*Lp*), small-worldness index, global efficiency (*Eg*), and local efficiency (*Eloc*). The *Cp* measures the extent of local interconnectivity or cliquishness of a graph [47]. The *Lp* between each possible pair of nodes is the number of edges in the shortest path between them, divided by all possible pairs of nodes in the network. It measures the extent of overall routing efficiency, where a high *Lp* represents less efficient information flow due to long routes [48]. The small-worldness index is the ratio between the normalized *Cp* and normalized *Lp*, which are standardized by dividing their values by random networks preserving the same degree distribution and connections with the original brain network. Those combining high local clustering with short paths, representing an effective system, show features of a small world network [48,49,50,51]. *Eg* is the average inverse shortest path length and captures the extent of information propagation in the network [52]. The *Eloc* is the local efficiency computed on node neighborhoods, describing the extent of information transfer of the respective node with all other nodes in the network [52].

### 2.7. Statistical Analysis of Network Properties

NBS analysis and network property calculations were conducted in the network analysis toolbox GRETNA (Version 2.0.0, http://www.nitrc.org/projects/gretna/ (accessed on 30 October 2020)), based on MATLAB, while the brain was visualized using the BrainNet Viewer toolkit (http://www.nitrc.org/projects/bnv/ (accessed on 30 October 2020)). When comparing the differences between global topological properties between two groups, a nonparametric permutation test [53] (permutation = 10,000) was used. Age and sex were entered as nuisance covariates to regress out any potential mixed effects [44]. In addition, we used false discovery rate (FDR) correction (at *q* = 0.05) to reduce false-positive errors caused by multiple comparisons [54,55].

## 3. Results

### 3.1. Demographic Characteristics

The study included 30 Cantonese-Mandarin BGs (23 women and 7 men, mean age = 21.00 years, standard deviation = 1.93 years) and 30 Mandarin MGs (21 women and 9 men, mean age = 21.43 years, standard deviation = 2.05 years). There were no significant differences in age, sex, or non-verbal IQ between the two groups, and the two groups had comparable educational and socioeconomic backgrounds. We also compared the pass rate of the CET-4 and College English Test Band 6 (CET-6, a national English level test higher than CET-4 in China) between the two groups and found no difference in English proficiency (Table 1).

### 3.2. Alterations of Functional Brain Connectivity in Bilinguals

The NBS approach identified two differentially interconnected subnetworks. One was more strongly connected in the BG group (component size = 69 edges, *p* < 0.0001, corrected), and the other was more strongly connected in the MG group (component size = 72 edges, *p* < 0.0001, corrected). We refer to the BG > MG subnetwork as simply the BG network and the MG > BG subnetwork as the MG network. The full BG (red) and MG (blue) networks involving the bilateral frontal, parietal, temporal, and occipital cortices are shown in Appendix A.

Since the number of edges in the subnetworks was large, visualization and explanation were challenging. To focus on brain regions with maximally different connectivity between groups, we identified nodes with a high sum of edge differences of at least 10, as in previous studies (Figure 1) [56,57]. The sum of the edges of a node was identified by counting the total number of its edges in both the BG and MG networks. This method enabled us to detect nodes that were most likely to be altered by a long-term BG experience. Table 2 lists the nodes that had a sum of at least 10 edges, including the right inferior parietal lobule (IPL), right postcentral gyrus (PoCG), right posterior cingulate gyrus (PCG), and left inferior frontal opercular part (IFGoperc). Figure 2 and Appendix A show these nodes as well as their functional partners in the BG and MG subnetworks, which have been selected for discussion.

### 3.3. Small-Worldness and Efficiency of Brain Functional Networks in Bilinguals

Figure 3 shows the global parameters between Cantonese-Mandarin BGs and Mandarin MGs. We found that there were no significant differences between the two groups for all global network metrics at all network sparsity thresholds, which means that there was no difference in network efficiency between the BGs and MGs. Both BGs and MGs had the small-worldness property in view of *γ* > 1 and *λ* ≈ 1, or *σ* > 1.

## 4. Discussion

In the current study, widespread differences in functional connectivity in bilateral hemispheres were found between Cantonese-Mandarin bilinguals (BGs) and Mandarin monolinguals (MGs). Compared to the Mandarin MG group, Cantonese-Mandarin BGs showed increased functional connectivity between the bilateral frontoparietal and temporal regions and decreased functional connectivity in the bilateral occipital cortex and between the right sensorimotor region and bilateral prefrontal cortex. In addition, no significant differences in network efficiency were found between the two groups.

### 4.1. Alterations of Functional Network Connectivity in the Cantonese-Mandarin Bilinguals

It is well-known that the left hemisphere is dominant for language. However, in the current study, we found that, compared with the Mandarin monolinguals, the significant differences of the functional connectivity were not only in the left hemisphere but also in the right hemisphere of the Cantonese-Mandarin bilinguals (see Figure 1), including the right inferior parietal lobule (IPL), postcentral gyrus (PoCG), and posterior cingulate gyrus (PCG) (see Figure 2). Two meta-analyses showed that the right hemisphere works in an inter-hemisphere manner during language processing, and mainly participates in lexical–semantic processing [58,59]. Hull et al. [60] found that bilinguals who acquired both languages before the age of six showed bilateral hemispheric involvement for processing both languages. Liu et al. [61] defined a language network for the Cantonese-Mandarin bilinguals who acquired a second language before the age of six and found that bilateral brain regions were involved. In this study, most of the Cantonese-Mandarin bilinguals acquired both languages before turning 6 years old, so the altered functional connectivity in bilateral frontal, parietal, temporal, and occipital cortices further confirmed the involvement of bilateral hemispheres in early bilinguals.

#### 4.1.1. Brain Networks with Stronger Functional Connectivity in Cantonese-Mandarin Bilinguals

First, we found that, compared with Mandarin MGs, Cantonese-Mandarin BGs had stronger functional connectivity in the frontoparietal regions (see Figure 1). Specifically, functional connectivity between the right posterior cingulate gyrus (PCG) and the left inferior frontal gyrus opercularis part (IFGoperc), left inferior frontal gyrus triangular part (IFGtriang), and bilateral supplement motor area (SMA) was stronger in the BGs compared with that in the MGs (Figure 2C); both the right inferior parietal lobule (IPL) and left IFG of BGs showed stronger functional connectivity with the left anterior cingulate gyri (ACG) and right medial superior frontal gyrus compared with MGs (see Figure 2A,D). The right PCG is a part of the default mode network (DMN), which is associated with internally oriented thoughts and protects the execution of long-term mental plans from distraction and from immediate environmental demands [62,63,64,65]. The left IFGoperc and IFGtriang are the classic left inferior frontal language regions that are particularly related to phonological processing [66,67], semantic decisions and choices [68], and control of interference from a nontarget language [66,67,69]. The right IPL participates in cognitive control [70,71] and language selection [72]. The SMA [73,74], left ACG [75,76,77], and right medial superior frontal gyrus [78] have been identified as important areas of inhibitory control in BG processing. Similar to this study, Grady et al. [10] found stronger functional connectivity in the DMN and between the bilateral frontal and parietal regions in alphabetic BGs than in MGs, which indicates that both alphabetic and logographic BGs are more dependent on the brain network of language control and inhibition than MGs.

Second, compared with Mandarin MGs, Cantonese-Mandarin BGs had stronger functional connectivity between the bilateral frontoparietal and temporal regions (see Figure 1). Specifically, both the right IPL and left IFG of the BGs showed stronger functional connectivity with the temporal regions, including the left middle temporal gyrus and bilateral hippocampus, compared with MGs (see Figure 2A,D). Additionally, functional connectivity between the right PCG and right Heschl gyrus (HES) was stronger in the BGs than in the MGs (see Figure 2C). The left IFG participates in phonological and semantic processing [79]. As a key component of the ventral stream that processes speech signals for comprehension, the left middle temporal gyrus plays an important role in phonological–semantic mapping and semantic information retrieval [62,80,81,82]. In addition, as a structure containing the human primary auditory cortex, the right HES is related to the processing of phonological information [83,84]. The bilateral hippocampus is mainly responsible for memory retrieval [79,85]. Therefore, we guessed that, compared with Mandarin MGs, Cantonese-Mandarin BGs were more dependent on the brain network of phonological and semantic processing and memory retrieval. In fact, using a seed-based approach, Luk et al. [9] also found bilinguals had stronger functional connectivity between the left IFG and the bilateral middle temporal gyrus and right IPL, and researchers attributed the more long-range frontal-parietal functional connectivity to the need for bilinguals to recruit a more distributed and bilateral functional network. In addition, task modality fMRI studies found that BGs showed increased activation in the middle temporal gyrus during semantic tasks compared with MGs [86,87]. Moreover, increased gray matter volume of the anterior temporal cortex, hippocampus [88,89], and HES [90] has been reported in BGs. Garcia-Penton et al. [16] reported that BGs had more structural connectivity between the left frontal and the parietal and temporal regions than MGs. Therefore, we believed that BG experience strengthened functional connectivity between the frontoparietal and temporal regions.

In summary, our results seemed to indicate that, compared with Mandarin MGs, Cantonese-Mandarin BGs had stronger functional connectivity in the brain network of language control, inhibition, phonological and semantic processing, and memory retrieval. Our correlation analysis suggested that stronger brain network connectivity in the BG network was related to the better interference inhibition control ability of bilinguals (see Appendix A), which supported our partial conjecture. Studies have indicated that when processing one language, BGs need to constantly select words in the target language and suppress interference from the nontarget language, which requires the participation of language control and inhibition [91,92,93,94]. In Guangdong Province, China, Cantonese mainly serves as a mother tongue acquired and used by listening and speaking. It is the main spoken language but is seldom used for reading and writing, since Mandarin is the official language. Cantonese and Mandarin share the same writing system but differ in phonology, lexicon, and grammar [26,27]. In this study, Cantonese-Mandarin BGs recruited from Guangdong Province are very proficient in speaking and understanding both languages (see Table 1). Therefore, compared with Mandarin MGs, no matter which language Cantonese-Mandarin BGs speak, there will be strong phonological and semantic interference from the other language. To control interference of the nontarget language, BGs need more participation of brain regions related to language control, inhibition, and phonological and semantic processing. As a result, Cantonese-Mandarin BGs developed stronger connectivity in the brain’s functional networks of language control, inhibition, phonological and semantic processing, and memory retrieval.

#### 4.1.2. Brain Networks with Weaker Functional Connectivity in Cantonese-Mandarin Bilinguals

First, compared with Mandarin MGs, Cantonese-Mandarin BGs showed weaker functional connectivity in the bilateral occipitoparietal cortex (see Figure 1). Specifically, compared with MGs, BGs had weaker functional connectivity between the right IPL and right inferior occipital gyrus (IOG) (see Figure 1) and between the right PCG and occipital regions, including the bilateral superior occipital gyrus (SOG) and IOG (see Figure 2C). The right IPL participates in cognitive control [70,71]. The bilateral IOG and SOG are related to high-level visual analysis and word recognition [95] and might engage in the visual phonological and semantic processing of Chinese characters [96,97]. Moreover, the left occipital region is more involved in feature analysis, while the right occipital region is more involved in holistic visual processing [98,99]. Therefore, our results seemed to indicate that, compared with Mandarin MGs, Cantonese-Mandarin BGs had weaker functional connectivity in visually processing written language. Both Cantonese and Mandarin are logographic languages and use the same writing system: Chinese characters. Chinese characters involve certain sequences of strokes with complex visual-spatial configurations, thus requiring great visual analysis of spatial information [100]. Previous studies found that visual analysis of Chinese characters engaged bilateral occipital regions [98,101]. In addition, Ma et al. [25] found that the visual word recognition of Mandarin involved more visuospatial analysis of the occipital cortex than that of Cantonese. In the Guangdong Province of China, as a mother tongue, Cantonese is mainly learned through listening and speaking, so it is a spoken language. However, all children begin to learn Mandarin in school. In school, they read and write in Mandarin, so Mandarin is only a teaching language for Cantonese-Mandarin BGs. However, for Mandarin MGs, Mandarin is not only a spoken language but also a teaching language. Therefore, compared with Mandarin MGs, Cantonese-Mandarin BGs are less proficient at obtaining the phonetic and semantic information of written words through visual analysis but are more adept at phonological analysis, which leads to weaker functional connectivity in the brain network of visual processing of a written language.

Second, compared with Mandarin MGs, Cantonese-Mandarin BGs showed weaker connectivity between the right postcentral gyrus (PoCG) and the bilateral prefrontal cortex (PFC) and right caudate nucleus (see Figure 2B). Specifically, compared with MGs, BGs had weaker functional connectivity between the right PoCG and the bilateral superior frontal gyrus (SFG), middle frontal gyrus (MFG), IFG, and right angular gyrus (ANG) (see Figure 2B). Tan et al. [102] found that a large neuroanatomical network involving the right PoCG, bilateral SFG, MFG, and IFG was activated during a visual phonology task of written Chinese. The dorsal lateral frontal system involving the bilateral SFG and MFG is responsible for the visuospatial analysis of Chinese characters and orthography-to-phonology mapping, and it is supposed to create a long-term storage center of phonological representations of Chinese words [23]. In addition, as a crucial region responsible for skilled Chinese reading, the left MFG is related to the representation and working memory of visuospatial information and cognitive resource coordination [97]. The right ANG was found to be involved in visual-spatial attention [103]. The ventral prefrontal system comprising the bilateral IFG contributes to the phonological processing of Chinese words [23,104] and might be involved in BG speech planning and production [72]. Moreover, the right PoCG is related to the sensorimotor control of the lips, tongue, jaw, and larynx [105,106], as well as speech production [107]. The right caudate nucleus is associated with speech production and sensory-to-motor coupling [108]. Therefore, our results seemed to indicate that, compared with Mandarin MGs, Cantonese-Mandarin BGs had weaker functional connectivity in the brain network of visual and phonology processing and speech production. The weak functional connectivity was evidently related to the less frequent use of Mandarin in daily life, according to the Hebbian learning rule that neurons working together through a great deal of experience would strengthen their connectivity [109,110,111]. The reduced frequency hypothesis indicates that words in each language used by BGs were less effectively used than the same words in the language used by MGs because MGs adopted the same language to express themselves, whereas BGs allocated their time between two languages [86]. Behavioral studies also showed that BGs had less efficient visual word recognition and speech production than MGs [25,112,113,114].

### 4.2. Small-Worldness and Efficiency of the Brain Functional Network in Bilinguals

Finally, the current study showed that both Cantonese-Mandarin BGs and Mandarin MGs had small-world properties. The small-world property means that the whole-brain functional network has high efficiency in information processing and transference [50]. Our results indicated that the long-term BG experience did not affect the efficiency of information processing of the whole-brain functional network. Until now, there has been no report on the difference in network efficiency of the whole-brain functional network between BGs and MGs. However, Garcia-Penton et al. [16] found that, compared with MGs, BGs had lower global efficiency in the whole-brain structural network. Although brain structural connectivity has been shown to be a physical substrate of resting-state functional connectivity [18,19], studies have found inconsistencies between the structural and functional networks [17,20]. We speculate that the acquisition of two languages from early childhood results in specialized brain functional subnetworks but has no influence on network efficiency for proficient BGs.

## 5. Limitations

There are several limitations in this study. First, since English education is universal in Chinese, all participants in this study had English learning experience. To minimize the potential confounding effects, we matched the English proficiencies of the two groups by recruiting subjects who had passed the College English Test Band 4 (CET-4, a national English level test in China). Second, the choice of brain region definitions in the AAL atlas template might affect brain network parameters and functional connectivity [115]. However, the AAL atlas template [38] has been widely used in brain network studies [39,40,41], thus ensuring that the results are comparable with previous studies. Third, the behavioral measures in the study were inadequate, making it hard to know what exactly the brain network differences mean. In the future, in accordance with the results, we will add more related behavioral measures to make this clear.

## 6. Conclusions

In conclusion, compared with Mandarin MGs, Cantonese-Mandarin BGs had stronger connectivity in the brain network related to language control, inhibition, phonological and semantic processing, and memory retrieval, whereas weaker connectivity in the brain network was related to visual and phonology processing and speech production. However, Cantonese-Mandarin BGs and Mandarin MGs had the same network efficiency in the whole-brain functional network. This study provides a global insight to brain functional network reorganization resulting from bilingual experience.

## Figures and Tables

**Figure 1 brainsci-11-00310-f001:**
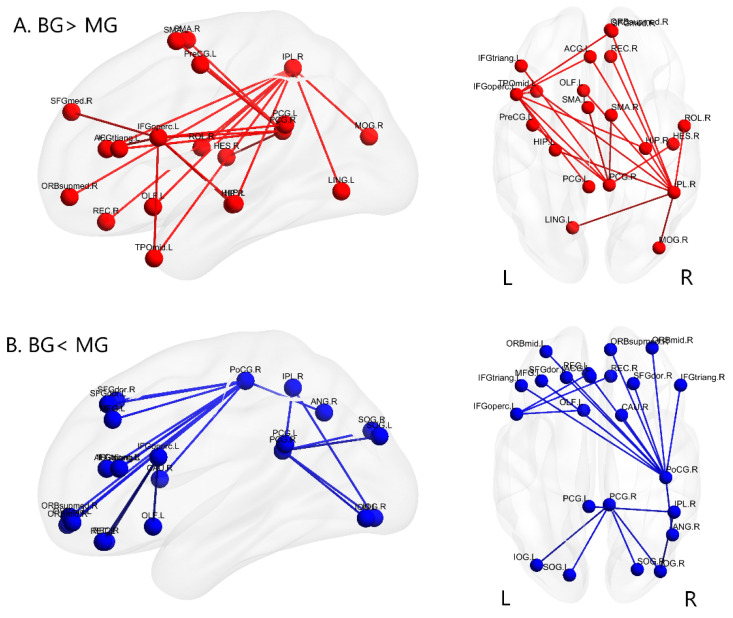
Whole-brain connectivity differences between the two groups. These components were thresholded for visualization to show nodes with a sum of edges differences ≥ 10 with all the nodes to which these suprathreshold nodes were connected. Red lines denote the subnetwork of bilinguals (BG) > monolinguals (MG) at *p* < 0.0001 after network-based statistics (NBS) correction (**A**). Blue lines denote the subnetwork of bilinguals (BG) < monolinguals (MG) at *p* < 0.0001 after NBS correction (**B**). Note that all figures are shown in neurological convention (subject’s left is the image’s left), and axial views are top-down.

**Figure 2 brainsci-11-00310-f002:**
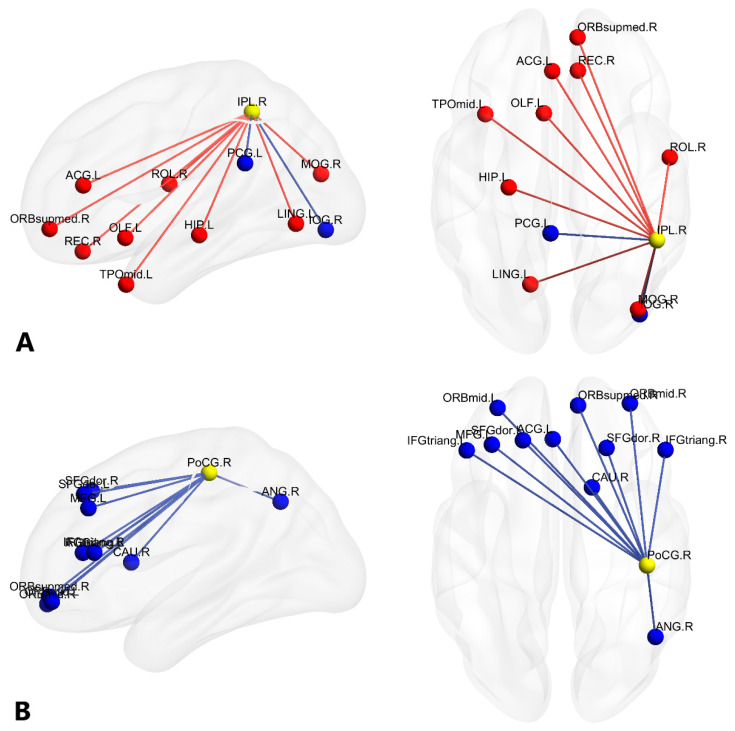
Visualization of selected nodes with the highest sum of edges in both the BG and MG subnetworks. In each figure, the selected node is shown in yellow (see Table 1 for the coordinates of selected nodes), while nodes more strongly connected to the selected node in the bilinguals (BG) > monolinguals (MG) (i.e., BG) network are shown in red, and nodes more strongly connected to the selected node in the MG > BG (i.e., MG) network are shown in blue (see Appendix A for the coordinates of all MG and BG partner nodes). Red and blue lines simply schematize connectivity between the selected node and each of its functional partners. (**A**) Connectivity between the right inferior parietal lobule (IPL) and all MG and BG partner nodes; (**B**) connectivity between the right postcentral gyrus (PoCG) and all MG and BG partner nodes; (**C**) connectivity between the right posterior cingulate gyrus (PCG) and all MG and BG partner nodes; (**D**) connectivity between the left inferior frontal opercular part (IFGoperc) and all MG and BG partner nodes. Note that all figures are shown in neurological convention (subject’s left is image’s left), and axial views are top-down.

**Figure 3 brainsci-11-00310-f003:**
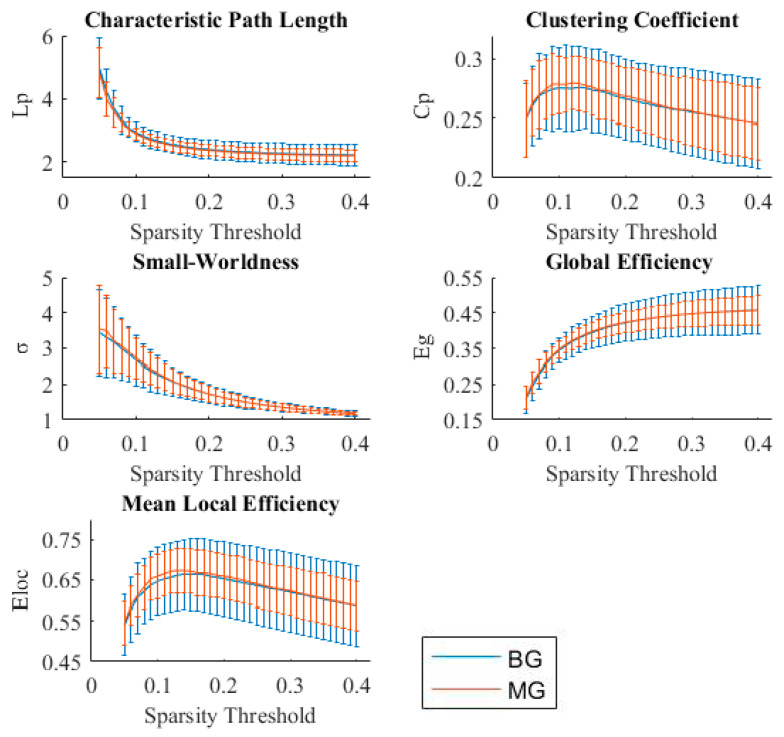
Comparison of global network metrics between Cantonese-Mandarin bilinguals and Mandarin monolinguals (BG = Cantonese-Mandarin bilinguals, and MG = Mandarin monolinguals).

**Table 1 brainsci-11-00310-t001:** Demographic characteristics and self-reported language assessment form the Language and Social Background Questionnaire (LBSQ) for participants in each group.

	BG	MG	*p* Value
N	30	30	--
Gender (male/female)	7/23	9/21	0.559 ^a^
Age (years, M ± SD)	21.00 ± 1.93	21.43 ± 2.05	0.402 ^b^
IQ (M ± SD)	122.50 ± 11.92	124.07 ± 14.10	0.644 ^b^
Education (undergraduate/postgraduate)	22/8	20/10	0.573 ^a^
District of residence (city/suburb)	20/10	19/11	0.787 ^a^
Father’s education			
Junior high school or below	13	12	0.957 ^a^
Senior high school or technical secondary school	7	7	
College or above	10	11	
Mother’s education			
Junior high school or below	11	14	0.542 ^a^
Senior high school or technical secondary school	7	8	
College or above	12	8	
Cantonese proficiency (M ± SD)			
Speaking^c^	9.40 ± 0.77	-	
Writing^c^	5.77 ± 2.13	-	
Understanding^c^	8.87 ± 0.94	-	
Reading^c^	7.87 ± 1.33	-	
Mandarin proficiency (M ± SD)			
Speaking^c^	8.53 ± 1.17	8.83 ± 1.18	0.326 ^b^
Writing^c^	8.67 ± 1.12	8.73 ± 1.26	0.829 ^b^
Understanding^c^	8.83 ± 1.12	8.77 ± 1.17	0.822 ^b^
Reading^c^	8.93 ± 0.98	9.00 ± 0.91	0.786 ^b^
English level (CET-4/CET-6)	14/16	13/17	0.795 ^a^

BG = Cantonese-Mandarin bilinguals; MG = Mandarin monolinguals; N = number of participants; M ± SD = mean ± standard deviation. ^a^ The *p* value was obtained using an *χ^2^*-test. ^b^ The *p* value was obtained using a two-sample *t*-test. ^c^ 0 = no proficiency; 10 = high proficiency.

**Table 2 brainsci-11-00310-t002:** Node-Level Analysis of Brain Connectivity Differences between Bilinguals and Monolinguals.

Node	Region	Coordinate (x, y, z)	BG > MG Edges	MG > BG Edges	Sum of Edges
A ^a^	IPL.R	(46.46, −46.29, 49.54)	9	2	11
B ^a^	PoCG.R	(41.43, −25.49, 52.55)	0	11	11
C ^a^	PCG.R	(7.44, −41.81, 21.87)	6	4	10
D ^a^	IFGoperc.L	(−48.43, 12.73, 19.02)	7	3	10

BG = bilinguals; IFGoperc = inferior frontal gyrus, opercular part; IPL = inferior parietal lobule; L = left; MG = monolinguals; PCG = posterior cingulate gyrus; PoCG = postcentral gyrus; R = right. ^a^ Shown in Figure 2.

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
