# Peer review of "The Differences in the Whole-Brain Functional Network between Cantonese-Mandarin Bilinguals and Mandarin Monolinguals"

_brainsci, 2021, doi:10.3390/brainsci11030310_

Round 1
Reviewer 1 Report
Review
This study entitled ''The differences in the whole-brain functional network between Cantonese-Mandarin bilinguals and Mandarin monolinguals’’ investigates whole-brain connectivity in bilingual individuals who master two logographic languages (Cantonese and Mandarin) versus monolingual individuals (Mandarin only).
The results of whole brain-connectivity differences without specific a priori hypotheses and behavioral measures to relate the brain differences with makes it hard to really know what the results mean exactly.
Introduction
The authors mention that most previous studies investigated bilinguals using alphabetic languages. In this study they investigate bilingual participants who speak two logographic languages. The introduction could be more elaborate in terms of what this study design will add to the field. Do the authors hypothesise that their study using two logographic language should yield different results about the bilingual brain or probably the results would replicate in a population of alphabetic bilinguals?
Also they argue that most previous studies used a seed-based approach while this study uses a whole-brain approach. Did the authors have more specific hypothesis other than ''In general, we expect to detect altered functional subnetworks in the BG brain for managing two languages’’. They should justify better why using a whole-brain approach and how it will answer specific questions still unanswered in the field of bilingualism and the brain.
Methods and Results
The Raven Progressive Matrices is a test for fluid intelligence of a non-verbal nature, since this study focuses on language it would have been relevant to have some measure of verbal IQ for the participants, would there be a way to get that information?
The only langue measure they have is the self-assessed language questionnaire. What about Education? And socio-economical status? It would be important to know to make sure the groups are equivalent and that the brain differences cannot be explained by other variables than bilingualism.
The bilingual participants learned their L2 between the ages of 3 and 7, since the critical period for learning an L2 is thought to be around 5 and 6 it would be interesting that they investigated whether there is a difference between the early learner and the late learners 5-6+.
Within the bilinguals is there a correlation between the AoA and brain connectivity measures?
Table 2 in the legend, what is meant by ''but supramarginal and angular gyri’’
''IPL, inferior parietal, but supramarginal and angular gyri’’
Since the self-rated proficiency is their only behavioral language measure. it would be interesing if they could extract the brain connectivity between their main ROIs and see if the connectivity relates to language proficiency in the participants.
Discussion
Line 329: ''stronger functional coupling in the frontoparietal region and between the frontoparietal and temporal regions could reflect better information integration of auditory phonological-semantic processing and inhibition control of this process in Cantonese-Mandarin BGs compared with Mandarin MGs.’’
I’m not sure their results really support the statement of better information integration of auditory phonological-semantic processing, could they relate this statement to the specific brain connectivity results that support this (which ROIs specifically).
In general, in the discussion the authors often fail to mention if they are talking about the left or right hemisphere ROIs. Especially when talking about language functions we know that the left and right brain regions are differently involved. Talking about the left parieto-frontal network should not be discussed the same way as the connectivity between the right parietal and the left frontal regions, the left and right parietal areas have different functions.
Section 4.1
Could the authors expand the discussion about the recruitment of bilateral hemispheres in bilingual individuals.
Line 281. ‘’Previous studies have confirmed that Chinese processing involves bilateral hemispheres.’’ Does Chinese processing involves the right hemisphere more than alphabetic language processing?
It has been suggested before that bilingualism involves the right hemisphere more (more bilateral processing) even for alphabetic languages, could they expand the discussion more regarding how their result relate to this? The authors cite the Hull and Vaid reference, but they don’t clearly relate their result with it. Does their result of increased connectivity in the right hemisphere relates more to the fact that it is Chinese or it relates to bilingualism?
Line 308, a lot of the connectivity results are, for example from a seed in the right hemisphere to regions of the left hemisphere. Ex: From the RIPL to the left cingulate or left hippocampus… how do they interpret that specifically since those are very distant areas and not directly connected to one another.
Did this study Grady, Luk, Craik and Bialystok [10] also find their differences in brain connectivity between areas in opposite hemispheres for the fronto-parietal network?
Line 313: ‘’The IPL participates in phonological and semantic processing’’, that’s classically for the left hemisphere, their results is with the Right IPL, could they expand more on the classical role of the R IPL? Do they think here that the RIPL becomes more involved in the language network or do they think its involvement has to do with its role for visual-spatial processing?
Again, between line 325 and 332, they relate their results to previous results and mention fronto-parietal coupling but is it really the same the connectivity between the right parietal area and the left frontal areas and other results showing increased connectivity within the left hemispheres? What does it mean to have greater connectivity between the R IPL and the left frontal lobe, there must be an intermediate area mediating the connectivity between those non-homologous bilateral brain areas.
Staring line 306, about the Temporal lobe connectivity result, increased connectivity between the Right IPL and the Left temporal pole and the R IPL and the Left hippocampus, those are not the classical language areas of the temporal lobe, the authors should be careful not to overinterpret their results by stating generalities about the temporal lobe. They should re-write part of the discussion and discuss their results more specifically and their interpretation.
The result of weaker functional connectivity in the brain network of visual processing is interesting and the interpretation makes the most sense, if the other results are hard to interpret the authors could focus on this one.
Limitations section:
English proficiency should go in the methods section, it is relevant and should not be mentioned only at the end of the paper.
Author Response
We sincerely thank the editor and the reviewer for their valuable comments. Please see the attachment for our point-to-point response.

Reviewer 2 Report
Thank you for giving me the opportunity to review your scientific manuscript.
I would like to begin by stating that it is my opinion that the methods used as part of this project are adequate and scientifically sound. The methods and results are well-described and I have only made a few minor suggestions for their improvement.
In this article, the authors have used Network Based Statistical analysis to demonstrate that Cantonese-Mandarin college students have greater functional connectivity within a subnetwork of brain regions that included the right inferior parietal lobule, right posterior cingulate gyrus the left inferior frontal gyrus, compared with a group of Mandarin monolingual college students. By contrast the authors observed greater connectivity in right postcentral gyrus in the monolingual group compared to the bilingual group.
I believe that this paper makes an important and significant contribution to the current neurolinguistic literature. The participant groups being studied have been well described and I believe that the authors carefully considered their choice of the Mandarin monolingual group as a control for their Cantonese-Mandarin bilinguals.
The authors importantly highlight some key considerations related to the fact that the bilingual group will likely only use Cantonese to converse rather than to read and write. I think that this is referred to appropriately in the introduction and discussion.
While I believe that there is a lot of important information to be gained from investigating resting functional connectivity differences between bilingual and monolingual participants, I would urge the authors to be more cautious when interpretating their results. While I found that there was a rationale present for each interpretation of the results within the discussion, I felt that some of the statements made, for example about bilinguals having stronger connectivity in the inhibitory control network, were too strong given the whole-brain approach used in this project.
I agree that one possible interpretation of increased fronto-parietal functional connectivity may be that the experience of Cantonese-Mandarin bilinguals has resulted in greater connectivity in their language control network (Green & Abutalebi 2013). However, unfortunately the authors have not reported any behavioural results within this manuscript to support that statement. In addition, regarding a change in the inhibitory control network as a whole, I would have predicted greater connectivity between subcortical regions as well (see for example, (DeLuca et al. 2020)), and it is unclear whether this was observed or not.
I would therefore recommend that the authors address this absence of behavioural data in their limitations section and consider toning down some of their conclusions (regarding differences between the inhibitory control, auditory phonological-semantic processing and visual phonology processing networks). The authors may also choose to provide alternative interpretations in their discussion to make it a little more balanced.
Minor points to address:
Abstract
I think the opening statement is too general and comes before the authors highlight that they are in fact working with logographic-logographic bilinguals. There have been brain imaging studies that have investigated functional connectivity in logographic bilinguals (for example see (Wang et al. 2020) who report resting state functional connectivity in Chinese-English bilinguals). I recommend the authors make this opening sentence clearer if they want to highlight novelty of their study.
I think the authors should be more specific in the first line of their results in the abstract. In its current form, the sentence is not specific enough to the current study. In addition, even if it is more specific, I don’t think this sentence should go first as it is mainly supported by Supplementary Figure 1. The sentences that follow are clearer as they specifically refer to the groups in the project.
Materials and Methods
Given the importance of describing the bilingual and monolingual participants linguistic backgrounds, I believe it would be ideal to include Table S1 within the main report, if that is possible.
I recommend that a few more key parameter details be added to the Data acquisition section. At present, there are some details missing so another group may struggle to replicate the sequences used. For example, what was the original resolution / voxel size of the fMRI acquisition prior to the resampling? It would also be useful to know the type of head coil that was used. I recommend that the authors see the Appendix from (Poldrack et al. 2008) for guidance, and ensure the recommended details are included.
In addition, I noticed that the authors have published a paper in 2020 (Ma et al., 2020, Brain Research). Can the authors please clarify whether the resting state data was collected in the same participants that were recruited for that project? If so, was it during the same MRI session and at what point during the session was the resting state fMRI acquisition collected? This would be important to know for replication purposes and should be added to the Data acquisition section.
The software packages used were run in MATLAB. Please can the authors also include information on the version of MATLAB they were using for clearer replication.
Please include the specific chapter or chapters used from the textbook on Applied Multiple Regression/Correlation Analysis for the Behavioral Sciences (reference 42 in manuscript).
Results and Figures
Figure 1 is clear and well-described in the legend. However, the figure would benefit from the addition of node names (as has been done for Fig 2). Alternatively, the authors may choose to add a table with node names, to make the results easier for readers to follow.
Can the authors confirm whether Supplementary Figure 1 shows all the nodes that have been selected within the union mask (as part of the network based statistical analysis)? Can this figure be accompanied by a Table or Matrix including all remaining node names? Readers will then be able to see which edges survived thresholding and subsequently which nodes have a sum of edges that were less than 10.
Thank you for the opportunity to review this manuscript. I hope that you find my comments constructive and useful.
References
DeLuca, V. et al., 2020. Understanding bilingual brain function and structure changes? U Bet! A Unified Bilingual Experience Trajectory model.
Green, D.W. & Abutalebi, J., 2013. Language control in bilinguals: The adaptive control hypothesis. Journal of cognitive psychology (Hove, England), 25(5), pp.515–530.
Poldrack, R.A. et al., 2008. Guidelines for reporting an fMRI study. NeuroImage, 40(2), pp.409–414.
Wang, R. et al., 2020. Functional and structural neuroplasticity associated with second language proficiency: An MRI study of Chinese-English bilinguals. Journal of Neurolinguistics, 56, p.100940.
Author Response

(The authors gave the same response as above.)

Round 2
Reviewer 1 Report
The authors significantly improved the paper and adressed adequately my comments and suggestions.
Minor point:
Second paragraph page 12. Was ''HES'' defined before in the paper?